# Holographic Encryption Applications Using Composite Orbital Angular Momentum Beams

**Nian Zhang** [1,2,3,4], **Baoxing Xiong** [1,2,3,4], **Xiang Zhang** [1,2,3,4] **and Xiao Yuan** [1,2,3,4,*]

1   School of Optoelectronic Science and Engineering, Soochow University, Suzhou 215006, China
2   Collaborative Innovation Center of Suzhou Nano Science and Technology, Soochow University, Suzhou 215006, China
3   Key Lab of Advanced Optical Manufacturing Technologies of Jiangsu Province, Suzhou 215006, China
4   Key Lab of Modern Optical Technologies of Education Ministry of China, Suzhou 215006, China
*   Correspondence: xyuan@suda.edu.cn

**Abstract:** Optical orbital angular momentum (OAM) holography has been developed and implemented as a vital method for optical encryption. However, OAM holography can only be encoded and decoded with an OAM beam, which limits the level of optical encryption. Here, composite OAM beams are introduced using a computer-generated hologram (CGH) for holographic encryption. The target image is encoded with composite helical mode indices, and the OAM holographic image can only be reconstructed under a specific illuminating composite OAM beam. The experimental results are consistent with the theoretical design and numerical simulations, verifying that composite OAM beams can provide a higher security level for optical holographic encryption. The proposed method can be used to enhance anti-counterfeiting applications, secure communication systems, and imaging systems.

**Keywords:** orbital angular momentum holography; composite vortex beams; holographic encryption





## 1. Introduction

Optical holography technology provides a vital strategy for reconstructing both the intensity and phase information of an object with computer-generated holograms (CGH), and has been widely used for optical tweezers [1,2], optical encryption [3], three-dimensional displays [4,5], and data storage [6]. For conventional holographic systems, the different physical properties of light, including wavelength, polarization, and incidence angle, have been explored and have been proven to carry independent information channels for optical holographic encryption [7–10]. Moreover, orbital angular momentum (OAM), as an essential physical dimension of light, has attracted considerable attention [11]. In general, OAM is characterized by a helical phase factor, $\exp(il\theta)$, where $l$ represents the topological charge number and $\theta$ indicates the azimuthal angle of a helical wave-front. OAM states can enhance information capacity due to its physically unlimited orthogonal helical modes and can be widely applied in optical communications [2,12], meta-surfaces [13], and so on.

Recently, OAM beams have been experimentally implemented for holographic encryption, including the field of linear optics and nonlinear optics [14,15]. Ruffato et al. made the first attempt to encode and decode information by designing CGH with OAM beams and phase singularities [16]. After that, OAM multiplexing holography for optical encryption was explored [17–20]. Unlike the work of Ruffato et al., OAM multiplexing holography preserves the doughnut intensity distribution of the OAM mode throughout the spatial-frequency transform by discrete sampling. However, current methods only encode a single OAM phase mode into the information, which limits the level of holographic encryption. Optical vortices with a superimposed helical phase have been explored in the past few years. Composite OAM beams are expected to enhance encryption capabilities

by superimposing different OAM modes. To the best of our knowledge, holography with different superimposed OAM modes has not yet been explored or implemented.

In this study, we conducted an experiment utilizing composite OAM beams for holographic encryption. In the experiment, we obtained good results without post-processing of the reconstructed images. Holograms of target images with composite OAM beams were obtained using the adaptive weighted Gerchberg-Saxton (WGS) algorithm. Our results show that OAM states with composite superimposed vortex beams provide higher security compared with those of the previous holography systems. For a specific composite OAM beam, the holographic image can be reconstructed clearly; otherwise, it can become a blurred image. We verify the feasibility of using such composite OAM beams using optical experiments and numerical simulations.

## 2. Principle and Methods

A schematic diagram of composite OAM phase holography is shown in Figure 1. An incident beam with the expected OAM beam illuminates the CGH phase pattern, and the holographic image can be reconstructed clearly. Conversely, if the CGH phase pattern is illuminated with the wrong beam, the noise becomes too high and blurs the image.

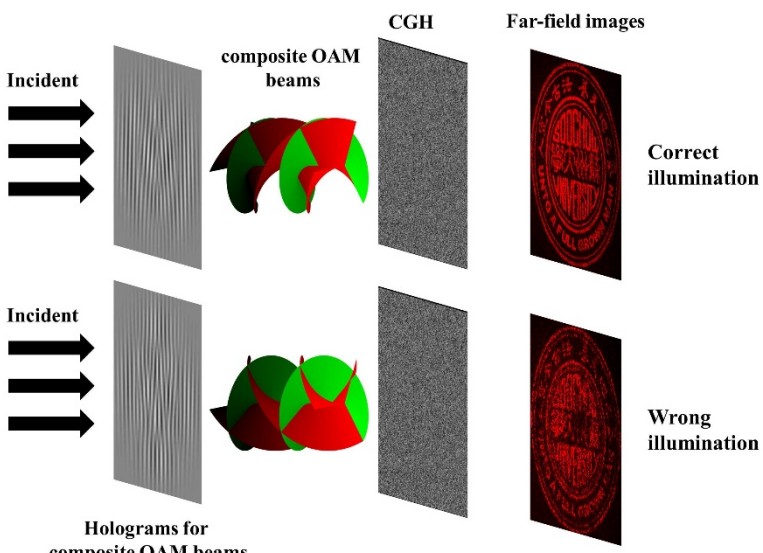

**Figure 1.** Schematic diagram of the composite OAM phase holography. Decoding of the CGH with the correct composite OAM beam: the encoding image appears in the far-field (the logo of Soochow University).

### 2.1. Composite-State Laguerre-Gaussian Beams

The complex amplitude of the Laguerre-Gaussian (LG) beam is written as [21]:

$$
\begin{aligned}
E(r,\theta,z) &= \sqrt{\frac{2p!}{\pi(p+|l|)!}}\,\frac{w_0}{w(z)}\left[\frac{r\sqrt{2}}{w(z)}\right]^{|l|}L_p^{|l|}\left[\frac{2r^2}{w^2(z)}\right]\exp\left[-\frac{r^2}{w^2(z)}\right]\exp\left[-\frac{ikr^2z}{2(z^2+z_R^2)}\right] \\
&\times \exp\left[i(2p+|l|+1)\tan^{-1}\frac{z}{z_R}\right]\exp(-il\theta)
\end{aligned}
\tag{1}
$$

where $w(z) = w_0\sqrt{1+(z/z_R)^2}$ is waist radius at the distance $z$, $w_0$ is the radius of the LG beam at the distance $z = 0$, $z_R = \pi w_0^2/\lambda$ is the Rayleigh length, $\lambda$ is the wavelength, and $k = 2\pi/\lambda$ is the wavenumber. $L_p^l(\cdot)$ represents the generalized Laguerre polynomial, whereas $p$ and $l$ represent the radial and angular mode numbers, respectively. The LG beam distribution possesses $(p+1)$ concentric rings. If set $p = 0$, each mode is a single ring.

Here, we discuss the situation where LG beams with a common waist radius at the source plane are coaxially superposed. The complex amplitude of the superposition of two LG beams of $(p_1, l_1)$ and $(p_2, l_2)$ can be written as:

$$E_{p_1,p_2}^{l_1,l_2}(r,\theta,z) = E_{p_1}^{l_1}(r,\theta,z) + E_{p_2}^{l_2}(r,\theta,z) \tag{2}$$

Next, we focus on the generation of a high-quality composite-state LG beam through a phase-only spatial light modulator (SLM). The vital step is to generate the desired hologram, which can be obtained by [22]:

$$\Phi_{SLM} = F_{amp} \cos\left[ Arg\left( E_{p_1,p_2}^{l_1,l_2} \right) + 2\pi f_x x \right] \tag{3}$$

where $F_{amp} = Abs(E_{p_1,p_2}^{l_1,l_2})$, and "*Abs*" and "*Arg*" are the amplitude and phase operations of the electric field of composite-state LG beams in Equation (2), respectively. A blazed grating with phase shift $2\pi f_x x$ ($f_x$ is the grating frequency) is used to control the diffraction angle of the light reflected from the phase-only SLM in the $x$ direction. Here, the relevant holograms of the composite OAM beams are plotted in Figure 2(a1–a5). Evidently, the holograms contain both amplitude and phase modulation, and the spatial phase-delay and grating depth determine the wave-front and intensity distribution of the incident light, respectively. Although most SLMs are phase-only devices, there are still several techniques to convert a phase-only device response into a full amplitude-phase response [23,24]. Therefore, in this experiment, the phase-only SLM was calibrated into a linear $2\pi$ phase response over all 256 gray level, and in this case, the residual amplitude modulation was negligible [25,26].

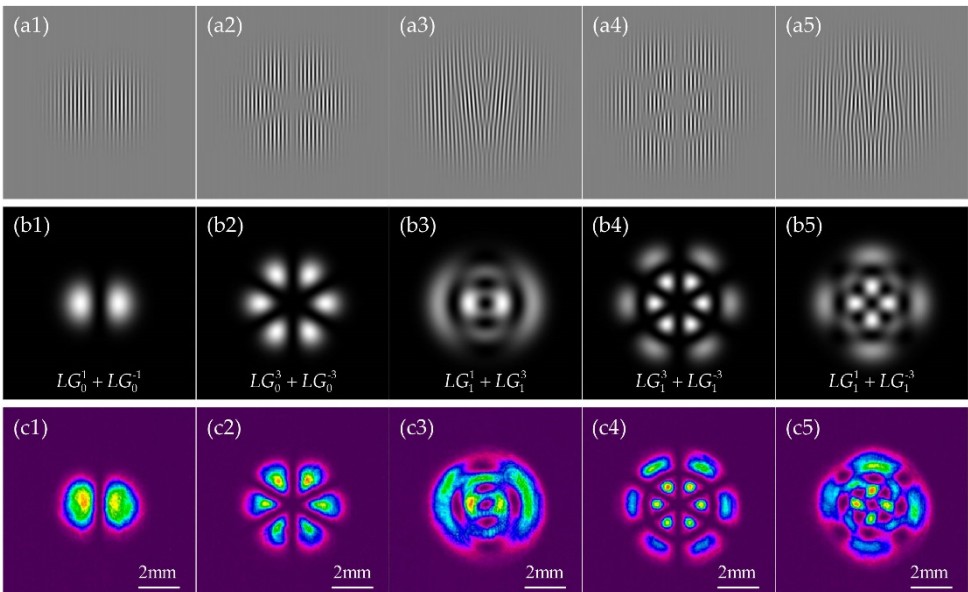

**Figure 2.** (**a1–a5**) The corresponding CGH phase patterns of composite OAM beams. (**b1–b5**) Simulated and (**c1–c5**) experimental intensity distributions of composite OAM beams.

The simulated and experimental intensity distributions of the composite OAM beams are shown in Figure 2(b1–b5,c1–c5), respectively. Note that in the case of $p_1 = p_2$ and $l_1 = -l_2$, we can observe the composite vortex beam composed of an optical bright-ring lattice containing $|l_1 - l_2|$ bright petals on each ring, and the helical phases disappear (OAM = 0), owing to the destructive interference of two LG beams with contrary topological charges. However, there are shear phase singularities, where the phase abruptly switches by $\pi$ [27,28]. In Figure 2(b3,b5,c3,c5), optical dark-ring lattices are observed where the phase singularities exist. The composite OAM beam has a dark vortex core with a topological

charge of 1 at the beam center, accompanied by three $|l_1\text{-}l_2|$ vortex singularities in the periphery.

The experimental results (Figure 2(c1–c5)) appear to be consistent with the numerical simulations (Figure 2(b1–b5)), and the slight differences are likely due to beam rotation while propagating. The beam rotation phenomenon can be explained from the perspective of the Gouy phase $\Phi = (2p + |l| + 1)arctg(z/z_R)$. As $p_1 = p_2$ and $l_1 = -l_2$, there is no rotation because both components of the beam have the same Gouy phase, which is also the case when the composite OAM beam exhibits no vortices [27,28]. As generation of the CGH phase pattern is influenced by the incident light field, we consider the propagation effect of the beam in the following section.

### 2.2. Design of the Composite-State LG Beam Hologram

An iterative Fourier transform algorithm is the best proper approach when designing an optimized CGH based on the forward and backward propagation of the light between the hologram plane and image plane [29,30]. We used a typical iterative Fourier transform algorithm (i.e., adaptive WGS algorithm) to suppress the speckle noise, reduce the appearance of artifacts, and improve the diffraction efficiency of the holographic image, based on Ref. [31].

Three different processed images with different levels of complexity were chosen, as shown in Figure 3a–c. The first image shows the letter "A", characterized by a grayscale channel (Figure 3a). The second image is the logo of Soochow University with pure black and white pixels (Figure 3b), and the third is a grayscale image of a tiger with finer details (Figure 3c). Figure 3d exhibits the flow chart of the adaptive WGS algorithm, with $\varphi_i$ and $\Phi_i$ representing phase distributions of the Fourier plane and the hologram plane in the $i$th iteration, respectively. The initial input field distribution $U_{OAM}^i$ is provided by the composite OAM beams. The initial phase value to start the iteration is [31]:

$$\varphi(x,y) = \exp[i2\pi(p\Delta X f_x + q\Delta Y f_y)] \tag{4}$$

where $(p, q)$, $(\Delta X, \Delta Y)$ and $(f_x, f_y)$ are the pixel coordinates, sampling intervals of the spatial domain, and frequency distribution in the $x$ and $y$ directions, respectively. During the iteration process, the amplitude constraint in the image plane is [31]:

$$A_{con} = \begin{cases} A_t \cdot W & \text{signal domain} \\ A_r & \text{noise domain} \end{cases} \tag{5}$$

where $W = \exp(A_t - A_r)$ is the weighting factor, and $A_t$ and $A_r$ are the target image amplitude and reconstructed image amplitude, respectively. With the progression of iterations, the normalized root mean squared error (*RMSE*) and peak signal-to-noise ratio (*PSNR*) are used to check the convergence of the algorithm. The RMSE and PSNR are defined as [31]:

$$RMSE = \sqrt{\frac{\sum\limits_{p=1}^{M}\sum\limits_{q=1}^{N}\left(A_t^2 - A_r^2\right)^2}{\sum\limits_{p=1}^{M}\sum\limits_{q=1}^{N}A_t^4}} \tag{6}$$

$$PSNR = 10log_{10}\left\{\frac{\left(2^b - 1\right)^2 MN}{\sum\limits_{p=1}^{M}\sum\limits_{q=1}^{N}\left(A_t - A_r\right)^2}\right\} \tag{7}$$

where $M$ and $N$ are the pixel numbers of the target image in $x$ and $y$ directions, respectively, and $b$ is the bit depth of images.

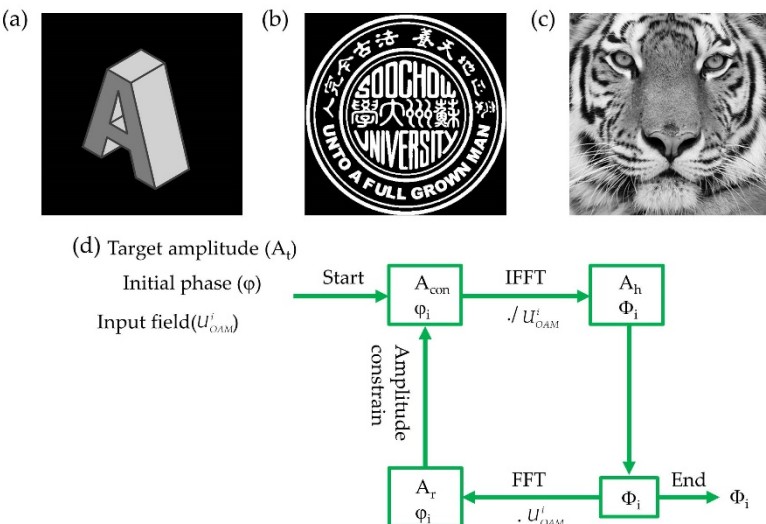

**Figure 3.** (**a**) Grayscale image of the letter "A"; (**b**) logo of Soochow University in pure white and black pixels, which includes the name and motto of the University in English and Chinese; (**c**) grayscale image of a tiger with finer details; (**d**) flow chart of WGS algorithm.

The simulation was performed to verify the feasibility of composite OAM phase holography, as shown in Figure 4. The intensity and phase distribution of the input composite OAM beam are shown in Figure 4a,b, respectively. The logo of Soochow University was used as a test image, partitioned into a signal and noise domain, and its size was $400 \times 400$, as shown in Figure 4c. The CGH of the logo is shown in Figure 4d. The wavelength of the laser and sampling intervals of the hologram were 632.8 nm and $12.5 \ \mu m \times 12.5 \ \mu m$, respectively. The focal length of the Fourier lens was 100 mm. As shown in Figure 4e,f, the simulation results showed the evolution of the RMSE and PSNR during WGS algorithm convergence to an optimized design of the CGH using a composite OAM beam with $(p_1, l_1) = (1, 1)$ and $(p_2, l_2) = (1, 3)$, indicating that encoded information in CGH with composite OAM beams can be an effective method for optical encryption.

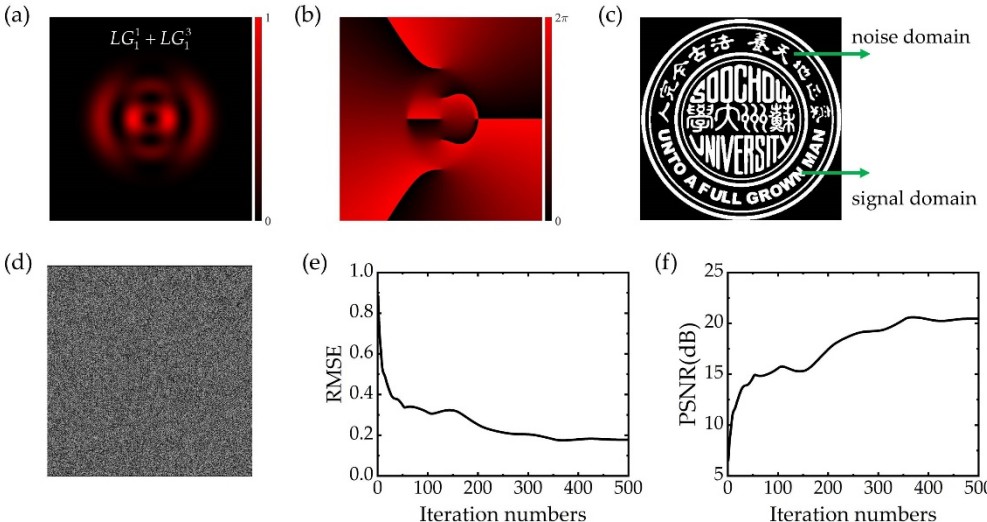

**Figure 4.** (**a**) The intensity and (**b**) phase distribution of the numerically simulated composite OAM beam; (**c**) target image containing signal and noise domain; (**d**) CGH of the target image (**c**); (**e**) RMSE; and (**f**) PSNR.

### 3. Experimental Results

The experimental setup of the composite OAM phase holography is shown in Figure 5. A He-Ne laser (Lumentum-1107P-632.8 nm-0.8 mW) was used to generate a Gaussian beam. The beam was expanded and collimated by a beam expander. An aperture was used to adjust the size of the beam irradiating on SLM1 (Holoeye-PLUTO-2-VIS-096, 1920 × 1080 pixels, pixel pitch of 8 μm). A polarizer was used to generate a P-polarized light. High quality composite OAM beams were generated by loading the corresponding CGH into SLM1. Then, the composite OAM beam was diffracted by SLM1 and irradiated onto SLM2 (Hamamatsu-X13138 series-07, 1272 × 1024 pixels, pixel pitch of 12.5 μm). Composite OAM phase holograms were generated by the WGS algorithm and loaded into SLM2. A CMOS (pco. edge-4.2 bi, 2048 × 2048 pixels, pixel pitch of 6.5 μm) was used to capture the images. To separate the zero point from the reconstructed image, a programmable zoom lens (PZL) ($f_{PZL}$ = 150 mm) was used, whose phase distribution is expressed as [32]:

$$\varphi_{PZL}(x, y) = -\frac{\pi}{\lambda f_{PZL}}(x^2 + y^2) \tag{8}$$

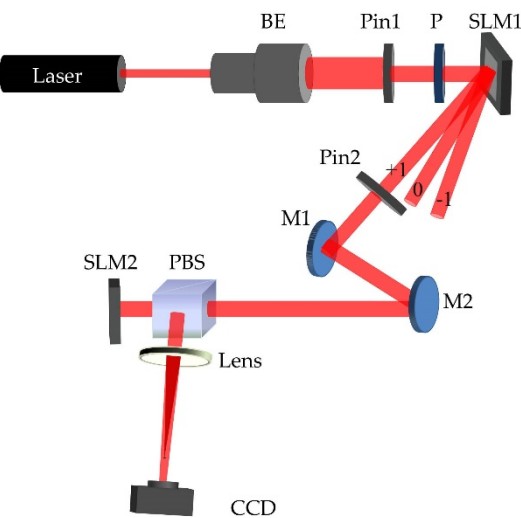

**Figure 5.** Schematic diagram of the experimental setup of composite OAM phase holography. BE: beam expander; Pin: pinhole aperture; P: linear polarizer; PBS: polarizing beam splitter; SLM: spatial light modulator.

Four different composite OAM phase hologram patterns were computed: $(p_1, l_1) = (0, 3)$ and $(p_2, l_2) = (0, -3)$ (Figure 6(a1–c1)), $(p_1, l_1) = (1, 1)$ and $(p_2, l_2) = (1, 3)$ (Figure 6(a2–c2)), $(p_1, l_1) = (1, 1)$ and $(p_2, l_2) = (1, -3)$ (Figure 6(a3–c3)), and $(p_1, l_1) = (1, 3)$ and $(p_2, l_2) = (1, -3)$ (Figure 6(a4–c4)). The simulated results for correct composite OAM beam illumination are shown in Figure 6a. Evidently, the reconstructed images will clearly appear for the correct beam illumination. The corresponding experimental results are shown in Figure 6b,c. For the correct beam illuminations, the experimental results (Figure 6b) were consistent with the numerical simulations (Figure 6a). However, when the holograms were irradiated by the wrong beam, the reconstructed images became blurred (Figure 6c).

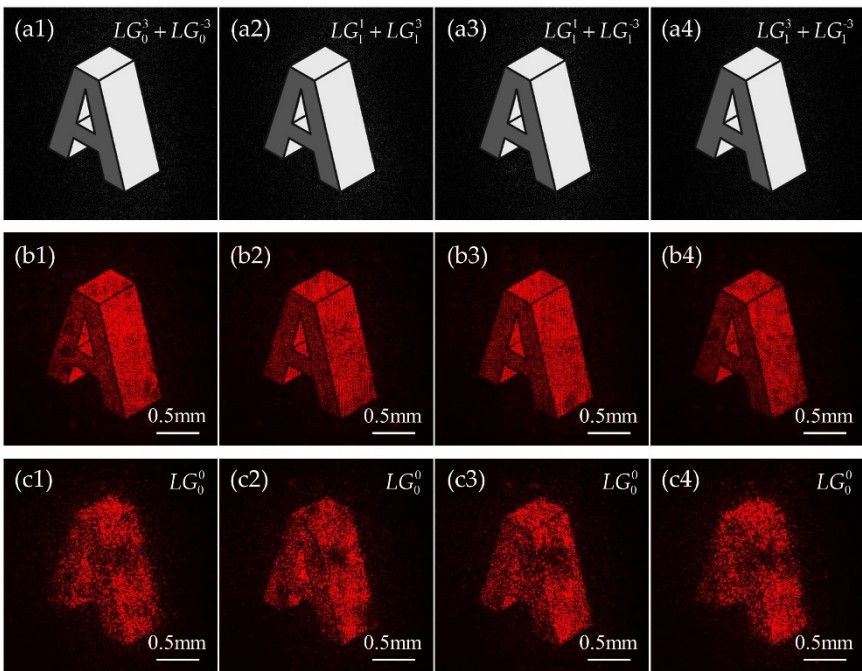

**Figure 6.** (**a**) Simulated results for correct illumination with (**a1**) $(p_1, l_1) = (0, 3)$ and $(p_2, l_2) = (0, -3)$, (**a2**) $(p_1, l_1) = (1, 1)$ and $(p_2, l_2) = (1, 3)$, (**a3**) $(p_1, l_1) = (1, 1)$ and $(p_2, l_2) = (1, -3)$, (**a4**) $(p_1, l_1) = (1, 3)$ and $(p_2, l_2) = (1, -3)$. (**b**) Experimental results for the composite OAM beam with (**b1**) $(p_1, l_1) = (0, 3)$ and $(p_2, l_2) = (0, -3)$, (**b2**) $(p_1, l_1) = (1, 1)$ and $(p_2, l_2) = (1, 3)$, (**b3**) $(p_1, l_1) = (1, 1)$ and $(p_2, l_2) = (1, -3)$, (**b4**) $(p_1, l_1) = (1, 3)$ and $(p_2, l_2) = (1, -3)$. (**c1**–**c4**) Experimental results with wrong Gaussian beam illumination of $(p, l) = (0, 0)$.

Then, the tiger image with finer details was encoded into the CGH phase pattern and computed for illumination with a composite OAM beam with $(p_1, l_1) = (1, 1)$ and $(p_2, l_2) = (1, 3)$. Several different composite OAM beams were used to test the optical response of the input composite OAM beams different from the optimal one, as shown in Figure 7. It is evident that for the correct composite OAM beam illumination, the finer details of the tiger are clearly displayed (Figure 7a). Conversely, when the wrong composite OAM beam illuminations are used, noise increases and the details of the image are no longer clearly visible (Figure 7b–d).

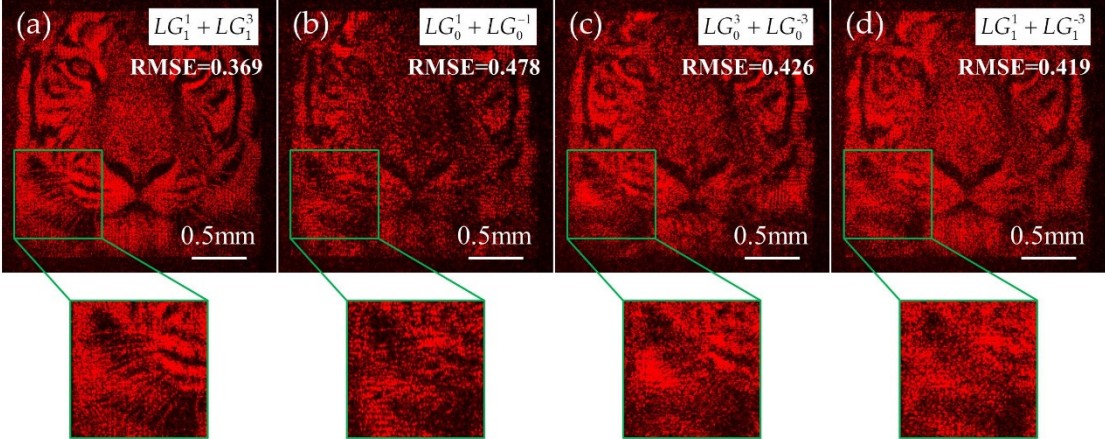

**Figure 7.** Experimental results of the CGH encoding grayscale image of a tiger. (**a**) The correct illumination of design was $(p_1, l_1) = (1, 1)$ and $(p_2, l_2) = (1, 3)$. The wrong illuminations used were (**b**) $(p_1, l_1) = (0, 1)$ and $(p_2, l_2) = (0, -1)$, (**c**) $(p_1, l_1) = (0, 3)$ and $(p_2, l_2) = (0, -3)$, and (**d**) $(p_1, l_1) = (1, 1)$ and $(p_2, l_2) = (1, -3)$.

As the calculated CGH phase pattern is illuminated by a specific intensity and phase distribution, the misalignment of the decoded beam with respect to the hologram position seems to affect the quality of the reconstructed image. To analyze the effects of displacement on the quality of the reconstructed image, the Soochow University logo was encoded into the CGH phase pattern and decoded using a composite OAM beam with $(p_1, l_1) = (1, 1)$ and $(p_2, l_2) = (1, 3)$. The CGH was illuminated with the correct composite OAM beam mode and size and was displaced the longitudinal and lateral directions of the beam. Moreover, because the composite OAM beam is axially symmetric, we moved the CGH along the positive $x$-axis direction. The phase distributions of the composite OAM beam with different lateral misplacements are shown in Figure 8(a1–a5). The change in the positions of the white circle indicates that the beam is being moved. As Figure 8(b1–b5) show, the reconstructed image gradually deteriorates and the details are no longer clear as the displacement increases in the lateral direction. The relatively high RMSs (~0.5) could have originated from the noise of the CCD or could from the uniformity of the incident beam.

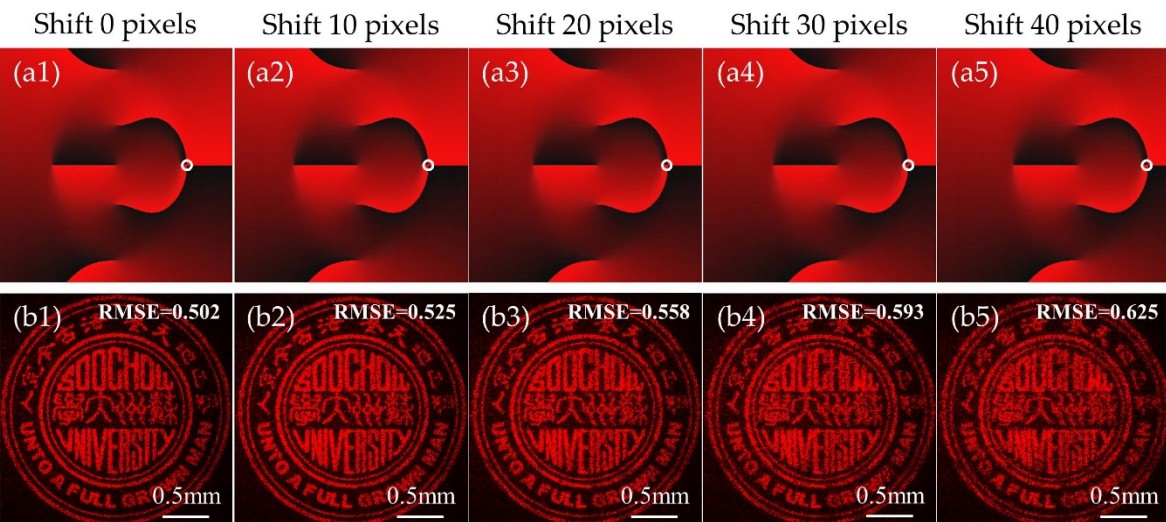

**Figure 8.** Experimental results increasing lateral misplacement of the Soochow University logo CGH with the composite OAM beam $(p_1, l_1) = (1, 1)$ and $(p_2, l_2) = (1, 3)$. (**a1–a5**) The phase distributions of different lateral misplacements, the white circle indicates the position of one of the phase singularities. (**b1–b5**) Experimental far-field images for different lateral misplacements. A pixel is 12.5 μm.

Furthermore, a parameter in the range of (0, 1] is represented by the different weights of the modes constituting the input beam. As shown in Figure 9(a1–a5,b1–b5), the different weights of the input beam result in different intensity distributions, but the phase distribution profiles are similar. From Figure 9(c1–c5), we can see that noise increases and the quality of the images decrease at smaller weights. The effect of the intensity distribution on the reconstructed image appears to be unimportant because the CGH is a phase-only holographic pattern, so the phase distribution of the beam plays a key role in the reconstructed image.

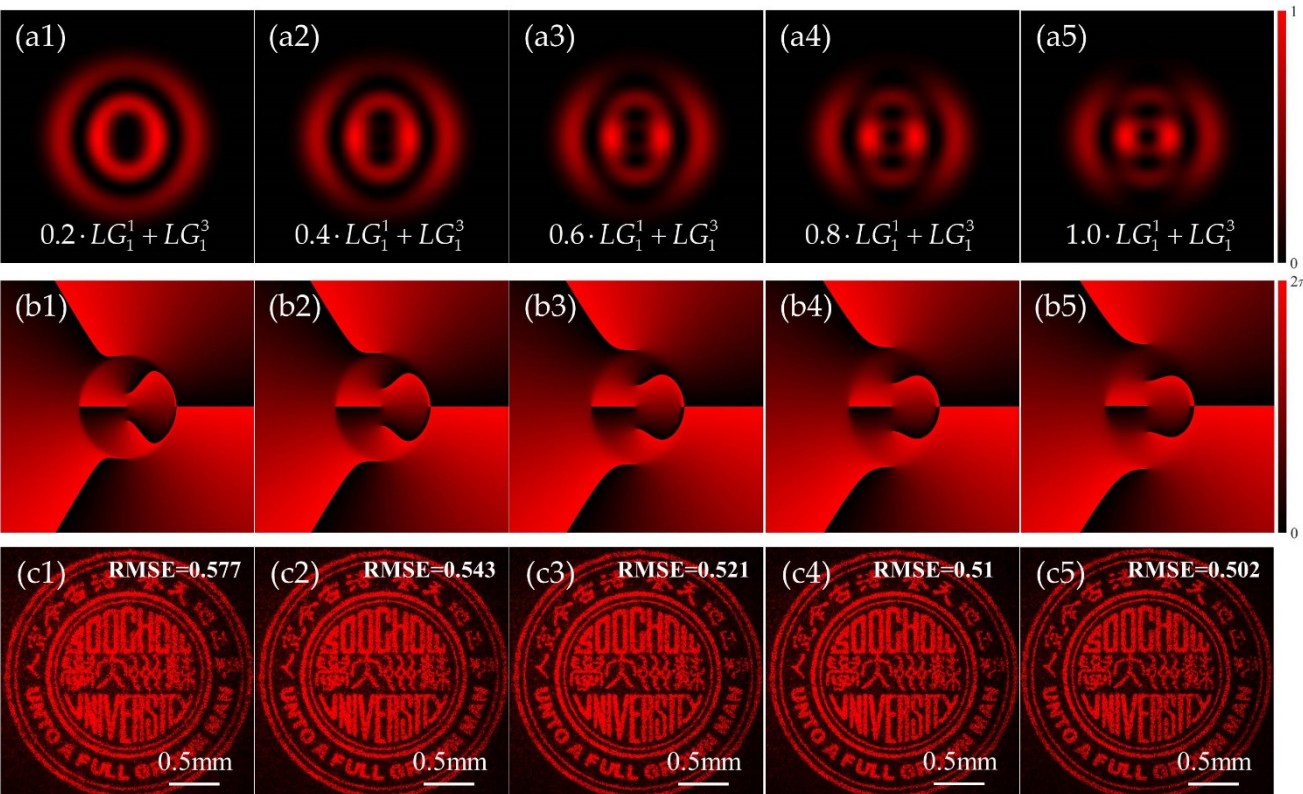

**Figure 9.** Analysis of the Soochow University logo CGH for different composite OAM beam energy ratios. (**a1**–**a5**) The intensity and (**b1**–**b5**) phase distributions of the composite OAM beams; (**c1**–**c5**) experimental far-field images for different beam energy ratios.

## 4. Discussion

We verified the feasibility of using composite OAM phase holograms to encode information through simulation and experimental results. In our experiment, we generated a high-quality composite OAM beam by loading the CGH into SLM1, and the angle between the incident light and the surface of SLM1 was less than 10° [33]. A circular aperture was used to choose the positive first-order diffracted light beam from SLM1. Using the described method, we could generate high-quality composite OAM beams with any radial and angular indices. Notaly, the waist size of the beam was a crucial parameter. In Ref. [34], for given beam indices, the image of the CGH encoding information could only be formed clearly in the neighborhood of the optimal beam waist. In contrast to beam indices, which can be represented as discrete integer values, the beam waist is a continuous parameter that can also be used as a dimension of the decoding key, further increasing security of the encoded information.

In this paper, we focused on the Fourier CGH, and the image was reconstructed by a Fourier lens. The misalignment of the lens introduced aberration distortion to the reconstructed image. However, further improvements may be considered. One method to achieve a lens-less reconstruction of a holographic image is by using the Fresnel CGH encoding. IAnother effective method is by adding the phase function of a Fourier transform lens on the Fourier CGH. In experiments, misalignment and beam energy ratio analyses can provide a good tolerance between the hologram and the phase distribution of the beam, and better images can be obtained will even micron-level precision. Hence, by designing composite OAM phase holographic experiments, information security can be improved and have wide application in the field of security and anti-counterfeiting.

## 5. Conclusions

In this paper, we present the detailed design and procedure of CGH information encoding for illumination with composite OAM beams. A high-quality composite OAM beam can be generated by loading the CGH into a phase-only SLM. The weighted Gerchberg-Saxton algorithm was implemented for the computation of the optimized CGH phase pattern. The calculation of the CGH for a given composite OAM beam resulted in a one-to-one correspondence between the holographic phase pattern and the beam, and the encoded information of CGH could be decoded with the correct illumination. The experimental results showed that the reconstructed images only clearly appeared; otherwise, the images appeared blurred. This proposed method has potential applications in anti-counterfeiting applications, secure communication systems, and imaging systems.

**Author Contributions:** Conceptualization, X.Y.; methodology, N.Z. and X.Y.; software, N.Z.; validation, N.Z. and B.X.; formal analysis, N.Z. and B.X.; investigation, N.Z.; resources, N.Z. and B.X.; data curation, N.Z.; writing—original draft preparation, N.Z.; writing—review and editing, X.Z. and X.Y.; visualization, N.Z.; supervision, X.Z. and X.Y.; project administration, X.Z. and X.Y.; funding acquisition, X.Z. and X.Y. All authors have read and agreed to the published version of the manuscript.

**Funding:** This research was funded by National Natural Science Foundation of China (NSFC) (61775153, 61705153), NSAF Joint Fund (U1930106), Natural Science Research of Jiangsu Higher Education Institutions of China (19KJA210001) and Priority Academic Program Development of Jiangsu Higher Education Institutions (PAPD).

**Institutional Review Board Statement:** Not applicable.

**Informed Consent Statement:** Not applicable.

**Data Availability Statement:** The available data has been stated in the article.

**Conflicts of Interest:** The authors declare no conflict of interest.

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
