# Peer review of "Holographic Encryption Applications Using Composite Orbital Angular Momentum Beams"

_photonics, doi:10.3390/photonics9090605_

Round 1

Reviewer 1 Report

In this manuscript, the authors proposed a holographic encryption method using composite orbital angular momentum beams. They designed the WGS algorithm to use composite OAM beams as reconstructive beams. Therefore, the correct images can only be decoded by the given OAM states. Although it is an interesting idea to use composite OAM beams as key in holographic encryption. Some important issues should be clarified before I can recommend the publication.

1.     The definition of the OAM holography in this paper is totally different from the previous works [Refs. 14, 15, 17, 18], I strongly suggest the authors to consider another expression to clarify the confusions.

2.     In Line 107, the authors mentioned “the WGS algorithm is implemented to improve the diffraction efficiency of the holographic image”, but I cannot find the quantitative analyzation of the diffraction efficiency.

3.     Actually, the concept of this paper is similar to Ref. [16]. In Line 46, The authors claimed the composite OAM states can enhance the security level. However, throught the manuscript, I cannot find the quantitative analyzation. Furthermore, a comparison with Ref. [16] is highly recommended.

Reviewer 2 Report

In this work, the authors experimentally showed that a phase hologram recorded using a reference composite OAM beam and reconstructed with another beam forms a noisy (smeared) image. All components of the work are known. The method for generating a composite OAM beam [21] using SLM is known [22–24], the method for calculating the phase of a hologram is described in [25]. Therefore, the novelty of the work is small and consists in the choice of a special reference beam for the hologram. But the fact that the composite OAM beam is more efficient than any other reference beam is not shown in this work. The work can be published after the authors take into account the comments.

Comments

1) In equation (1), it is necessary to multiply the function of the LG beam amplitude by the waist radius w0 so that the amplitude is dimensionless and less in modulus 1. Also, the azimuthal index of the Laguerre polynomial must be written in modulo, since it must always be positive.

2) Why is the amplitude function (3) given? To generate a composite OAM beam (2) using only phase SLM, the authors use the simplest coding method, ignoring the beam amplitude (2). The paper should give an exact expression for the phase transmission function of the SLM.

3) The paper considers composite OAM beams, but for beams (a2) and (a4) in Fig. 2, OAM is equal to zero.

4) Beams with OAM, which do not have circular symmetry, rotate during propagation. But the authors in line 101 incorrectly explain this rotation. They write that in the case of p1=p2 and l1=-l2, the beam does not rotate, since "two LG beams have opposite Gouy phae value". But it's not like that. The beams do not rotate since p1=p2. See the beam rotation conditions in Opt. Las. Eng. 29, 343 (1998).

5) Figure 3d does not show the reference OAM beam.

6) The adaptive iterative algorithm described in Section 2.2 is taken from [25]. But the adaptive modification of the Gerchberg-Saxton algorithm has been known for a long time. For example, see Optik, 88(1), 17 (1991). Iterative pink-ponk algorithms are known to not converge and stagnate. Therefore, the RMS error in Fig. 4f is too large - 20%.

7) In this work, it is shown that the image reconstructed from the hologram is distorted if the reconstructing beam differs from the recording beam. But there are no numbers. Authors should add data to the work as the RMS error changes in Figures 7-9.

8) Equation for RMS error (6) differs from standard. In the standard definition of RMS error, the summation should be carried out separately in the numerators of the intensity difference, and in the denominators of the intensity only.

9) In the experimental pictures, the authors should indicate the physical size of the frames.

Round 2

Reviewer 1 Report

Thanks for the efforts from the authors. 

Author Response

Thanks for your help and comments concerning our manuscript (photonics-1838736).

Reviewer 2 Report

The authors wrote very lengthy replies, but almost did not change the manuscript itself. Therefore, I repeat my remarks and ask the authors to make references to all the indicated articles and to make all the indicated changes to the text of the work. And highlight all the changes in the work with color so that it can be seen. Otherwise, I do not recommend publishing this work.

Comments

1) In equation (1), it is necessary to multiply the function of the LG beam amplitude by the waist radius w0 so that the amplitude is dimensionless and less in modulus 1.

2) The phase function (3) varies from -1 to 1, although the phase on the SLM can vary from 0 to 2pi. Encoding method 22 when field amplitude (2) is translated into phase (3) incorrectly. Therefore, the authors have to improve the SLM phase with the help of iterations.

3) The paper considers composite OAM beams, but for beams (a2) and (a4) in Fig. 2, OAM is equal to zero.

4) Beams with OAM, which do not have circular symmetry, rotate during propagation. But the authors in line 101 incorrectly explain this rotation. They write that in the case of p1=p2 and l1=-l2, the beam does not rotate, since "two LG beams have opposite Gouy phae value". But it's not like that. The beams do not rotate since p1=p2. See the beam rotation conditions in Opt. Las. Eng. 29, 343 (1998).

5) Figure 3d does not show the reference OAM beam.

6) The adaptive iterative algorithm described in Section 2.2 is taken from [25]. But the adaptive modification of the Gerchberg-Saxton algorithm has been known for a long time. For example, see Optik, 88(1), 17 (1991). Iterative pink-ponk algorithms are known to not converge and stagnate. Therefore, the RMS error in Fig. 4f is too large - 20%.

7) In this work, it is shown that the image reconstructed from the hologram is distorted if the reconstructing beam differs from the recording beam. But there are no numbers. Authors should add data to the work as the RMS error changes in Figures 7-9.
